# The Impact of Preconception Gastric Bypass Surgery on Maternal Micronutrient Status before and during Pregnancy: A Retrospective Cohort Study in the Netherlands between 2009 and 2019

**DOI:** 10.3390/nu14040736

**Published:** 2022-02-09

**Authors:** Katinka Snoek, Nadia van de Woestijne, Sten Willemsen, René Klaassen, Sander Galjaard, Joop Laven, Régine Steegers-Theunissen, Sam Schoenmakers

**Affiliations:** 1Department of Obstetrics and Gynecology, Erasmus MC, University Medical Center, 3000 CA Rotterdam, The Netherlands; k.snoek.1@erasmusmc.nl (K.S.); nadia.woestijne@gmail.com (N.v.d.W.); s.willemsen@erasmusmc.nl (S.W.); s.galjaard@erasmusmc.nl (S.G.); r.steegers@erasmusmc.nl (R.S.-T.); 2Department of Biostatistics, Erasmus MC, University Medical Center, 3000 CA Rotterdam, The Netherlands; 3Department of Epidemiology, Erasmus MC, University Medical Center, 3000 CA Rotterdam, The Netherlands; 4Department of Surgery, Maasstad Hospital, 3079 DZ Rotterdam, The Netherlands; klaassenr@maasstadziekenhuis.nl; 5Division of Reproductive Endocrinology and Infertility, Department of Obstetrics and Gynecology, Erasmus MC, University Medical Center, 3000 CA Rotterdam, The Netherlands; j.laven@erasmusmc.nl

**Keywords:** gastric bypass, vitamin status, malnutrition, supplements, treatments, pregnancy, periconception, micronutrients

## Abstract

Post-bariatric weight loss can cause iatrogenic malnutrition and micronutrient depletion. In this study, we evaluated the impact of gastric bypass surgery (GB) and multivitamin supplement use on maternal micronutrient status before and across pregnancy. A retrospective medical chart review of 197 singleton pregnancies after GB with a due date between 2009 and 2019 was performed at a bariatric expertise center in the Netherlands. Hemoglobin, calcium, iron status, folate, vitamin D, vitamin B12 and ferritin levels were determined before and after GB during standard follow-up and at all gestational trimesters and analyzed using linear mixed models. Patients were prescribed standard multivitamin supplements or multivitamins specifically developed for post-bariatric patients (FitForMe WLS Forte (FFM)). Overall, hemoglobin and calcium levels decreased after surgery and during pregnancy, whereas folate, vitamin D, and vitamin B12 levels increased, and iron levels remained stable. FFM use was associated with higher hemoglobin, folate, vitamin D, and ferritin levels. In conclusion, through adequate supplementation and follow-up, GB does not have to result in impaired micronutrient status. Supplements developed specifically for post-bariatric patients generally result in higher micronutrient values than regular multivitamins before and during pregnancy. These data emphasize the urgent need for nutritional counseling including dietary and multivitamin supplement advise for post-bariatric women contemplating and during pregnancy.

## 1. Introduction

The global obesity incidence has reached pandemic proportions, and its related morbidity and mortality places a heavy burden on society and individuals, including women of reproductive age. The best solution to diminish obesity is adopting healthier nutritional behaviors in combination with a healthy lifestyle to accomplish weight loss. Nonetheless, only fewer than 1% of women with obesity are successful in achieving and maintaining a normal weight, and long-term adherence to dietary interventions is generally low [1].

Bariatric surgery (BS) is envisioned as a durable solution for severe obesity as it leads to substantial and permanent weight loss. Internationally, patients qualify for this treatment if they have a body mass index (BMI) above 40 kg/m^2^ or a BMI above 35 kg/m^2^ with obesity-related comorbidities such as hypertension [2]. Although obesity rates are similar between men and women, 73.7% of post-bariatric patients are female, of which most are of reproductive age [3,4].

Traditionally, three types of BS are performed with different postsurgical anatomical effects. The types of BS include restrictive surgery, which decreases stomach capacity and limits the amount of food intake (e.g., sleeve gastrectomy); malabsorptive surgery, which reduces nutrient uptake by bypassing the small intestines and limiting food absorption; and a combination of the two (e.g., gastric bypass surgery).

Despite reducing obesity-related complications, BS can also increase the risk for iatrogenic malnutrition due to postoperative malabsorption and excessive, rapid weight loss. Post-bariatric micronutrient deficiencies during pregnancy occur in up to 70%, usually due to inadequate dietary intake and decreased absorption of nutrients [5,6,7,8,9,10].

During the periconception period, defined as 14 weeks before until 10 weeks after conception, appropriate micronutrient levels are essential for the maturation of oocytes and embryonic, fetal, and placental development [11]. For example, folate deficiencies can result in reduced oocyte quality, subfertility, and impaired fetal development, whereas vitamin B12 deficiencies are associated with neural tube defects, miscarriage, and preterm birth [12,13,14,15]. Pregnancy is characterized by metabolic and endocrine adaptations and increased needs for nutrients such as iron and folate to meet the increased demand needed for maternal health and placental and fetal development.

In order for possible post-bariatric micronutrient deficiencies to be treated and prevented, it is advised that supplements be prescribed postsurgically and that micronutrient levels be monitored on a regular basis [16]. Besides the risk for deficiencies, excess micronutrient supplementation can also have detrimental consequences for both mother and child [17,18]. Consensus recommendations for periconception care of these patients have been proposed [19]. However, international guidelines regarding treatment and prevention of suboptimal micronutrient levels in post-bariatric pregnant women are lacking, as are post-bariatric pregnancy-tailored multivitamin supplements.

With this retrospective observational cohort study in a post-bariatric population, we aimed to determine the course of maternal micronutrient status before and during pregnancy in women who have undergone gastric bypass surgery. Secondarily, we also evaluated the micronutrient status of these patients using standard multivitamin supplements in comparison to those using post-bariatric supplements.

## 2. Materials and Methods

### 2.1. Study Design and Participants

We performed a retrospective observational cohort study. Pregnancies after gastric bypass surgery with a due date between January 2009 and December 2019 were included. Women were included at the bariatric expertise center of the Maasstad Hospital, the Netherlands, using diagnosis treatment combination (DTC) codes for BS and any pregnancy-related DTC code after BS. An additional search was performed using the appointment code referring to patients who visited the bariatric outpatient clinic for follow-up during pregnancy. 43.7% of the patients were women of reproductive age (between 18 and 45 years old).

We included both first and consecutive pregnancies after gastric bypass surgery. Gestational age was determined by crown-rump length between 10^+0^ and 12^+6^ weeks of gestation after the last menstrual period, or by conception date plus 14 days after in vitro fertilization. Twin pregnancies and pregnancies that did not occur after gastric bypass surgery were excluded.

The combination of DTC codes for BS and pregnancy and a search for the appointment code of bariatric patients with a pregnancy resulted in 206 post-bariatric pregnancies. There were 3 twin pregnancies, which were excluded from the analysis. Since the introduction of BS in the last century, different techniques have been practiced and preferred over the years. Due to the results (e.g., complication rate, outcomes) of the different types of BS, currently gastric bypass surgery is the most used and practiced BS technique. Therefore, only pregnancies after gastric bypass surgery were included in the study group (*n* = 197).

#### 2.1.1. Gastric Bypass Surgery

During this surgery, the stomach is divided into a small pouch using a linear stapler. After that, a Roux-en-Y construction is created, consisting of an alimentary limb of 150 cm and a biliopancreatic limb of 50 cm (Figure 1).

#### 2.1.2. Clinical Parameters

General patient characteristics including age at conception, type of BS, obesity-related comorbidities such as hypertension and diabetes mellitus, multivitamin supplement use, blood pressure, length (cm), weight (kg) and BMI (kg/m^2^) before and after surgery were registered.

#### 2.1.3. Multivitamin Supplement Use

After surgery, patients were supplemented to prevent micronutrient deficiencies. The Maasstad Hospital advises the life-long use of the multivitamin WLS Forte by FitForMe^®^ (Rotterdam, The Netherlands) (FFM), specifically developed for post-bariatric patients worldwide [20]. Vitamin concentrations in FFM are dosed above the daily recommended dose in order to correct for malabsorption after BS [20].

#### 2.1.4. Laboratory Blood Analysis

Serum levels regarding hemoglobin, calcium, iron status, folate, vitamin D, vitamin B12 and ferritin were routinely collected once preoperatively and postoperatively at 6, 12, and 18 months and thereafter yearly as clinical practice according to standard local follow-up protocols. Moreover, these levels were determined in all gestational trimesters. These serum levels were measured according to the local post-bariatric protocol directly after the blood draw. They provide valuable information about the micronutrient status and can also influence fetal health during pregnancy.

#### 2.1.5. Statistical Analysis

Descriptive statistics were used for patient characteristics, which were tested for normal distribution. Continuous, normally distributed variables were presented as mean with standard deviation, and variables with a skewed distribution as median with interquartile range (IQR). Categorical variables were presented as counts and proportions. To estimate the effect of gastric bypass surgery on the levels of the micronutrients and hemoglobin, we first used multiple imputation to create 10 datasets in which the missing values were substituted with likely non-missing values. Rubin’s rules were used to pool the results from the imputation sets. Besides the outcome, we also used the covariates BMI (at blood draw or before surgery), season of the year, ferric carboxymaltose infusions, vitamin B12 injections, diabetes mellitus, ethnicity, smoking, educational level, age, time interval between blood draw and BS, first versus consecutive pregnancy, and multivitamin use. The effect of season on micronutrient status was modelled through a cyclic B-spline with 3 degrees of freedom, the first component peaking at the end of December/beginning of January, the second at the end of March/beginning of April, and the last at the end of June/beginning of July (with ‘autumn’ being the reference).

Next, a mixed model was estimated with the micronutrients as outcome of interest. We used two random intercepts: one at the level of the mother and one at the level of the pregnancy, in order to take into account the correlation in the response between observations. Note that for purposes of modelling the correlation structure, each of the observations that were not in a pregnancy, (i.e., the measurements before surgery and those after surgery but before pregnancy) were considered to constitute a ‘pregnancy’ of their own with only a single measurement.

From the effect of the mixed models, we estimated the contrast between the measurements before gastric bypass surgery and each of the other measurements (preconceptionally after surgery, and during the first, second, and third trimesters of pregnancy).

To estimate the effects of vitamin supplementation, we proceeded in a similar manner. Additionally, we included the type of vitamin supplement as a covariate and replaced the covariate BMI before surgery with BMI at the time of blood draw. Hereby, we incorporated the effect that gastric bypass surgery may have on BMI in the adjustment. There was no correction for multiple testing.

A *p*-value <0.05 was considered statistically significant. Statistical analyses were performed using SPSS (IBM SPSS Statistics Version 25) and R version 4.

## 3. Results

Table 1 describes the baseline characteristics of the study group. The median BMI before surgery was 43.3 kg/m^2^ (IQR 40.9–46.2), which decreased to 29.0 kg/m^2^ (IQR 26.6–32.5) periconceptionally (*p* < 0.001). Median time between gastric bypass surgery and conception was 18.7 months (IQR 10.2–31.3). Median age at conception was 29.8 years (IQR 26.3–33.4). Most patients had a Caucasian background (63.2%). Of the 197 pregnancies, the pregnancy outcomes were as follows: miscarriage or induced abortion (*n* = 38), intra-uterine fetal demise (*n* = 1), live birth (*n* = 121), or missing data due to loss of follow-up (*n* = 37).

### Micronutrient Blood Levels before Surgery, after Surgery, and during Pregnancy

The estimated means of hemoglobin and micronutrient levels are shown in Table 2. Hemoglobin and calcium levels gradually decreased before and during pregnancy, whereas folate levels increased after surgery (from 12.0 nmol/L before surgery to 27.5–31.7 nmol/L during pregnancy). Iron levels also increased after surgery (from 12.6 µmol/L before surgery to 14.9 µmol/L after surgery preconceptionally), and vitamin B12 and vitamin D levels generally increased as well. Ferritin levels increased after surgery, while they seemed to decrease during pregnancy. A total of 36.9% of the patients (*n* = 69) were administered additional vitamin B12 injections during pregnancy on an individual basis according to vitamin B12 status. Figure 2a–g show the individual serum levels per pregnancy preconceptionally before and after surgery and during pregnancy.

Importantly, there was no significant association between first versus consecutive pregnancies and hemoglobin and micronutrient status (*p*-value all >0.05).

The associations between gastric bypass surgery, hemoglobin, and micronutrient concentrations after surgery pre- and postconceptionally, are shown in Table 3, having been corrected for the covariates. Gastric bypass surgery was found to be associated with lower hemoglobin levels during the second trimester (β 0.701, 95% CI 0.515–0.887) and the third trimester (β 0.906, 95% CI 0.709–1.103). Calcium serum levels were all significantly lower after surgery, whereas folate levels were significantly higher. The effect on vitamin B12 levels was less consistent (after surgery and during the first trimester, vitamin B12 levels were significantly higher, whereas they were significantly lower during the second and third trimesters). Gastric bypass surgery had no significant effect on vitamin D and ferritin levels. Intraclass correlation coefficients showed that most of the variance was residual and cannot be explained by the pregnancy or the woman. The numbers were consistent along the micronutrient values (mostly regarding the pregnancies, Appendix A).

The association between gastric bypass surgery, hemoglobin, and micronutrient concentrations in (1) FFM versus other multivitamin supplementation (preconceptionally after surgery) and (2) FFM versus standard pregnancy multivitamin (during pregnancy) is shown in Table 4. Appendix A shows the dosages of prescribed supplements in the study group. During the first and third trimesters, hemoglobin was significantly higher in the FFM group. Folate levels after surgery, vitamin D after surgery and from the second trimester, and ferritin during the first trimester were higher in the FFM group. However, vitamin B12 levels were significantly lower in the FFM group. Significantly fewer FFM users were treated with vitamin B12 injections during pregnancy compared to users of standard multivitamins (18.7 vs. 53.5%, *p* < 0.001).

## 4. Discussion

Maternal micronutrient status, both preconceptionally and during pregnancy, is associated with fetal and pregnancy outcome [17,18]. BS is known to diminish vitamin and nutrient absorption due to the anatomical changes resulting in malabsorptive postsurgical effects. Regarding the altered anatomy after gastric bypass surgery and the sites of absorption of micronutrients, the risks for decreased vitamin B12, vitamin D, and iron levels are expected to increase postoperatively (Figure 1). Close monitoring and long-term follow up of the nutritional status after surgery is therefore warranted.

This study aimed to investigate the course of maternal micronutrient status preconceptionally and during pregnancy in women after gastric bypass surgery. We also evaluated the micronutrient status postoperatively with the use of standard multivitamin supplements compared to supplement prescription aimed specifically at post-bariatric patients (FFM).

We demonstrate that gastric bypass surgery does not have to result in impaired micronutrient status if supplementation and follow-up are adequate. FFM was associated with higher hemoglobin, folate, vitamin D, and ferritin serum levels preconceptionally and during pregnancy. Additionally, consecutive pregnancies showed no significant effect on micronutrient values compared with first post-bariatric pregnancies, indicating effects of bariatric surgery itself instead of time period-related effects after gastric bypass surgery on pregnancy.

### 4.1. Postoperative Effects of Supplementation on Vitamin Status

An adequate folate status has shown to be vital for fetal health [21]. Our results show that using supplements specifically aimed at post-bariatric women (FFM) results in improved folate status in women using FFM, as serum folate levels were significantly higher. In addition, iron status increased in our post-bariatric population, independent of the choice of supplementation. Higher oral iron supplementation dosage is known to decrease the proportion of absorbed iron and result in gastrointestinal side effects. Therefore, increased oral supplementation does not appear as feasible [22,23]. If impairment of the iron status is noted, despite supplement usage, we suggest intravenous supplementation with extra attention to iron status and proper supplementation to create an optimal starting situation before achieving a pregnancy.

After gastric bypass surgery, vitamin B12 deficiencies are common and are prevented in clinical practice by vitamin B12 supplementation (oral or intramuscular) [5,6,7,8,9]. This study shows that vitamin B12 levels can increase after BS if supplementation is adequate. The higher maternal vitamin B12 status in the standard multivitamin group, in contrast with vitamin B12 being dosed higher in FFM, can be explained by vitamin B12 injections being significantly more often administered. Therefore, another benefit of the usage of FFM is the reduced need for intramuscular vitamin B12 injections.

Medeiros et al., reported that 72% of patients after gastric bypass surgery had low first trimester vitamin D levels, despite daily supplementation of 15 µg vitamin D [24]. Our patients used either FFM (containing 75 µg vitamin D) or regular multivitamins (containing 10 µg vitamin D), and the FFM group showed significantly higher vitamin D levels. These findings indicate that post-bariatric women generally need more than the regular 10–15 µg vitamin D supplementation per day to be supplemented properly. Moreover, skin color and skin exposure to sun can affect vitamin D status, and therefore healthcare professionals should be extra aware of the risk of vitamin D deficiencies in patients with darker skin and with less skin exposure to sun (e.g., during the winter season).

Our results show that daily multivitamin supplementation; adequate follow-up; close monitoring; and timely supplementation, including seasonally additional supplementation, can prevent the occurrence of these deficiencies. The first trimester is the most vulnerable window for embryonic and fetal development, and micronutrient deficiencies during this stage are associated with adverse pregnancy outcome such as prematurity, miscarriage, preeclampsia, and growth restriction [25,26,27]. Hence, it is imperative to monitor these micronutrient blood levels and treat them accordingly, preferably before conception.

### 4.2. Multivitamin Supplementation Adherence

The low percentage of decreased micronutrient levels in our study may reflect the effect of adequate follow-up and routine micronutrient monitoring. Nausea and vomiting are common symptoms during pregnancy, especially during the first trimester. As a result, the prescribed vitamin supplements could either be not fully absorbed or the adherence to these supplements could be lower. Previous research shows that adherence to supplements is only 30% at 6 months after BS and decreases over time, and another study found that only 34.7% of post-bariatric pregnant women adhered to supplementation [28,29]. Hazart et al. showed that counseling on vitamin supplementation increased adherence from 23.5% after surgery to 77.8–100% during pregnancy [9]. In conclusion, educating post-bariatric women by explaining the vital role of micronutrient status appears a promising strategy to prevent deficiencies during the periconception period and beyond [9].

### 4.3. Clinical Implications

Micronutrient status can be addressed by a dietetic review to optimize the diet, aside from only prescribing vitamin supplements [30]. Consecutive post-bariatric pregnancies did not show a significant effect on micronutrient status, indicating that gastric bypass surgery entails a long-lasting effect on micronutrient status, one that does not seem to improve with time. Therefore, consecutive post-bariatric pregnancies should not be regarded as lower-risk pregnancies compared to first post-bariatric pregnancies. This study does not support the advice to wait 18 months between surgery and conception since micronutrient status will not improve over time.

Unplanned pregnancies occur more often after BS due to the restoration of the menstrual cycle after weight loss [31]. Consequently, post-bariatric women may not have prepared optimally for a pregnancy by taking folic acid before conception [32]. Therefore, post-bariatric women of reproductive age have to be made aware of the increased chance of achieving an (un)wanted pregnancy, and adequate intake of folic acid should be emphasized.

### 4.4. Research Implications

Future research should focus on trimester-dependent supplementation and strategies to further improve supplementation adherence. This study shows that a one-size-fits-all approach does not apply to post-bariatric women, and individual treatment adjustments are necessary, taking the patient’s preferences into account.

### 4.5. Strengths and Limitations

Strengths of this study are the high number of pregnancies included after gastric bypass surgery, combined with not only longitudinal clinical follow-up starting before surgery, but also follow-up of first and consecutive pregnancies including longitudinal blood collections. As this study focused on gastric bypass surgery, healthcare professionals can provide more tailored information and education for this specific patient group.

A limitation of this study is the maximum measurable upper limit of serum folate levels of the used assay at the Maasstad Hospital of 45.3 nmol/L. Given the high incidence of this level, many pregnancies are likely to have folate serum levels above this limit. Moreover, despite the well-organized follow-up of the study group, loss of follow-up was inevitable, resulting in missing data.

## 5. Conclusions

If supplementation and follow-up are adequate, preconception gastric bypass surgery is not associated with an impaired micronutrient status. This study does not confirm the advice to wait 18 months before becoming pregnant after gastric bypass surgery; however, healthcare professionals do have to make women aware of the increased chance of becoming pregnant in order to prepare them optimally. Our study confirms the importance of adequate periconception nutritional care, including multivitamin supplementation and monitoring of the micronutrient status before and during pregnancy, with a clinical multidisciplinary team for post-bariatric women.

## Figures and Tables

**Figure 1 nutrients-14-00736-f001:**
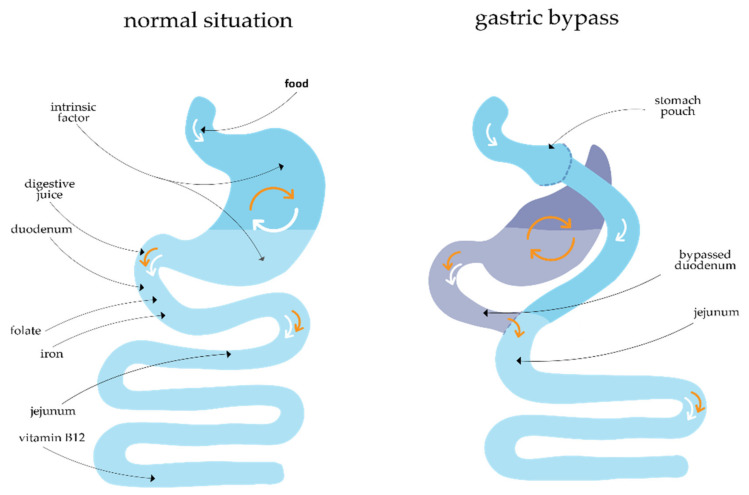
Normal situation versus altered situation after gastric bypass surgery.

**Figure 2 nutrients-14-00736-f002:**
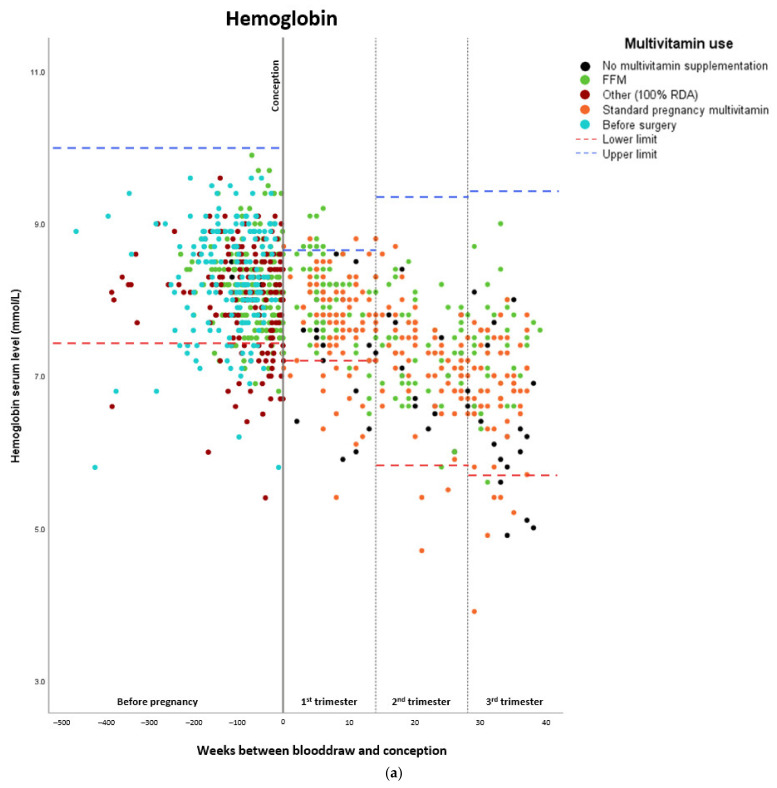
Maternal hemoglobin and micronutrient status before and during pregnancy, the Netherlands, 2009-2019. All values between the lower and upper limits are within the normal range. (**a**) Maternal hemoglobin status. (**b**) Maternal folate status. * 45.3 nmol/L is the upper measurable limit for folate. (**c**) Maternal iron status. (**d**) Maternal vitamin B12 status. (**e**) Maternal vitamin D status. (**f**) Maternal calcium status. (**g**) Maternal ferritin status.

**Table 1 nutrients-14-00736-t001:** Baseline characteristics of the included post-gastric bypass pregnancies (*n* = 197), the Netherlands, 2009–2019.

Characteristic	Study Population
Age at conception in years	
Median	29.8 (26.6–33.4)
Missing	0
Parity	
Nulliparous	78 (39.6)
Missing	0
Educational level	
Low	33 (17.5)
Intermediate	122 (58.7)
High	45 (23.8)
Missing	9
Comorbidity	
Diabetes mellitus	11 (5.6)
Hypertension	13 (6.7)
Obstructive sleep apnea syndrome	2 (1.0)
Asthma/COPD	30 (15.4)
Hypothyroidism	24 (12.3)
Missing	2
Period between surgery and conception in months	18.7 (10.2–31.3)
Missing	16
BMI before surgery (kg/m^2^) *	
Median	43.3 (40.9–46.2)
Missing	0
BMI at conception (kg/m^2^)	
Median	29.0 (26.6–32.5)
Missing	44
Geographic origin	
Caucasian	122 (63.2)
African	12 (6.2)
Asian	1 (0.5)
Other/mixed	58 (30.1)
Missing	4

Descriptive statistics were used to describe patient characteristics, and variables were tested for normal distribution. Numbers are presented as *n* (valid %) or median (IQR). * BMI = body mass index.

**Table 2 nutrients-14-00736-t002:** Estimated marginal means of hemoglobin and micronutrient levels before gastric bypass surgery, after surgery, and during pregnancy, the Netherlands, 2009–2019.

**Hemoglobin**				
	Estimated marginal means (mmol/L)	Standard error	Lower confidence interval (2.5%)	Upper confidence interval (97.5%)
Before surgery	8.30	0.059	8.184	8.416
Preconceptionally after surgery	8.10	0.062	7.979	8.221
First trimester	7.79	0.057	7.677	7.903
Second trimester	7.21	0.060	7.092	7.328
Third trimester	6.99	0.063	6.868	7.113
**Folate**				
	Estimated marginal means (nmol/L)	Standard error	Lower confidence interval (2.5%)	Upper confidence interval (97.5%)
Before surgery	12.0	1.13	9.785	14.215
Preconceptionally after surgery	22.0	1.14	19.766	24.234
First trimester	31.7	1.11	29.524	33.876
Second trimester	31.7	1.13	29.485	33.915
Third trimester	27.5	1.33	24.893	30.107
**Calcium**				
	Estimated marginal means (mmol/L)	Standard error	Lower confidence interval (2.5%)	Upper confidence interval (97.5%)
Before surgery	2.38	0.009	2.363	2.397
Preconceptionally after surgery	2.33	0.010	2.311	2.349
First trimester	2.30	0.010	2.281	2.319
Second trimester	2.24	0.009	2.222	2.258
Third trimester	2.23	0.013	2.205	2.255
**Iron**				
	Estimated marginal means (µmol/L)	Standard error	Lower confidence interval (2.5%)	Upper confidence interval (97.5%)
Before surgery	12.6	0.588	11.448	13.752
Preconceptionally after surgery	14.9	0.629	13.667	16.133
First trimester	16.1	0.583	14.957	17.243
Second trimester	14.7	0.630	13.465	15.935
Third trimester	14.2	0.737	12.755	15.645
**Vitamin B12 ***				
	Estimated marginal means (pmol/L)	Standard error	Lower confidence interval (2.5%)	Upper confidence interval (97.5%)
Before surgery	257.24	1.056	230.442	287.149
Preconceptionally after surgery	357.81	1.065	316.494	404.518
First trimester	327.01	1.058	292.790	365.236
Second trimester	290.04	1.058	259.885	323.681
Third trimester	320.54	1.062	284.919	360.610
**Vitamin D**				
	Estimated marginal means (nmol/L)	Standard error	Lower confidence interval (2.5%)	Upper confidence interval (97.5%)
Before surgery	33.0	1.9	29.276	36.724
Preconceptionally after surgery	53.2	1.99	49.300	57.100
First trimester	49.3	1.86	45.654	52.946
Second trimester	48.4	1.96	44.558	52.242
Third trimester	48.4	2.15	44.186	52.614
**Ferritin**				
	Estimated marginal means (µg/L)	Standard error	Lower confidence interval (2.5%)	Upper confidence interval (97.5%)
Before surgery	50.6	6.05	38.742	62.458
Preconceptionally after surgery	68.2	6.39	55.676	80.724
First trimester	55.7	5.85	44.234	67.166
Second trimester	37.0	6.06	25.122	48.878
Third trimester	34.3	7.29	20.012	48.588

* All micronutrient values represent absolute differences, whereas vitamin B12 values represent relative differences. For vitamin B12, a difference of 0.05 corresponds with a difference of 5%. Linear mixed models were used, unadjusted model.

**Table 3 nutrients-14-00736-t003:** The associations between gastric bypass surgery, hemoglobin, and micronutrient concentrations after gastric bypass surgery before and during pregnancy, the Netherlands, 2009–2019. The timepoint ‘before surgery’ is used as reference category.

**Hemoglobin**				
	Beta	Standard error	Lower confidence interval (2.5%)	Upper confidence interval (97.5%)
Preconceptionally after surgery	0.069	0.072	–0.072	0.211
First trimester	0.171	0.088	–0.001	0.342
Second trimester	0.701	0.095	0.515	0.887
Third trimester	0.906	0.101	0.709	1.103
**Folate**				
	Beta	Standard error	Lower confidence interval (2.5%)	Upper confidence interval (97.5%)
Preconceptionally after surgery	–9.087	1.501	–12.029	–6.145
First trimester	–18.872	1.754	–22.311	–15.434
Second trimester	–18.893	1.937	–22.688	–15.097
Third trimester	–14.783	2.071	–18.842	–10.724
**Calcium**				
	Beta	Standard error	Lower confidence interval (2.5%)	Upper confidence interval (97.5%)
Preconceptionally after surgery	0.037	0.012	0.014	0.061
First trimester	0.057	0.013	0.030	0.083
Second trimester	0.110	0.016	0.079	0.141
Third trimester	0.122	0.017	0.088	0.156
**Iron**				
	Beta	Standard error	Lower confidence interval (2.5%)	Upper confidence interval (97.5%)
Preconceptionally after surgery	–2.027	0.867	–3.727	–0.327
First trimester	–3.313	0.983	–5.240	–1.385
Second trimester	–1.744	1.130	–3.959	0.471
Third trimester	–1.570	1.182	–3.887	0.748
**Vitamin B12 ***				
	Beta	Standard error	Lower confidence interval (2.5%)	Upper confidence interval (97.5%)
Preconceptionally after surgery	–0.245	0.086	–2.093	–1.123
First trimester	–0.072	0.104	–0.748	0.356
Second trimester	0.087	0.107	–0.334	0.807
Third trimester	0.009	0.111	–0.568	0.753
**Vitamin D**				
	Beta	Standard error	Lower confidence interval (2.5%)	Upper confidence interval (97.5%)
Preconceptionally after surgery	–2.179	2.473	–7.026	2.668
First trimester	–2.118	3.002	–8.001	3.766
Second trimester	2.046	3.275	–4.372	8.464
Third trimester	–2.108	3.458	–8.884	4.669
**Ferritin**				
	Beta	Standard error	Lower confidence interval (2.5%)	Upper confidence interval (97.5%)
Preconceptionally after surgery	–25.684	8.546	–42.434	–8.935
First trimester	–18.306	9.998	–37.902	1.291
Second trimester	–5.117	10.666	–26.023	15.789
Third trimester	–1.045	11.424	–23.436	21.346


* All micronutrient values represent absolute differences, whereas vitamin B12 values represent relative differences. For vitamin B12, a difference of 0.05 corresponds with a difference of 5%. Linear mixed models were used, adjusted for season, ferric carboxymaltose infusions, vitamin B12 injections, diabetes mellitus, ethnicity, smoking, educational level, age at blood draw, time interval between blood draw and bariatric surgery, first versus consecutive pregnancy, BMI before surgery, and multivitamin use.Vitamin Supplements: FitForMe WLS Forte^®^ versus Standard Supplements

**Table 4 nutrients-14-00736-t004:** The associations between gastric bypass surgery, hemoglobin, and micronutrient concentrations in (1) FitForMe versus other multivitamin supplementation (after surgery before pregnancy) and (2) FitForMe versus standard pregnancy multivitamin (during pregnancy), the Netherlands, 2009–2019. The FitForMe group is used as a reference category.

**Hemoglobin**				
	Beta	Standard error	Lower confidence interval (2.5%)	Upper confidence interval (97.5%)
Preconceptionally after surgery	0.106	0.103	–0.096	0.309
First trimester	0.213	0.105	0.006	0.420
Second trimester	0.031	0.114	–0.194	0.255
Third trimester	0.426	0.123	0.184	0.667
**Folate**				
	Beta	Standard error	Lower confidence interval (2.5%)	Upper confidence interval (97.5%)
Preconceptionally after surgery	6.953	2.117	2.803	11.103
First trimester	3.179	2.353	–1.433	7.791
Second trimester	1.038	2.360	–3.587	5.663
Third trimester	3.230	2.577	–1.821	8.281
**Calcium**				
	Beta	Standard error	Lower confidence interval (2.5%)	Upper confidence interval (97.5%)
Preconceptionally after surgery	0.006	0.020	–0.032	0.045
First trimester	–0.002	0.021	–0.043	0.038
Second trimester	–0.007	0.023	–0.052	0.038
Third trimester	–0.023	0.024	–0.070	0.025
**Iron**				
	Beta	Standard error	Lower confidence interval (2.5%)	Upper confidence interval (97.5%)
Preconceptionally after surgery	0.674	1.243	–1.763	3.112
First trimester	1.839	1.215	–0.543	4.221
Second trimester	1.597	1.306	–0.963	4.157
Third trimester	1.557	1.613	–1.605	4.719
**Vitamin B12 ***				
	Beta	Standard error	Lower confidence interval (2.5%)	Upper confidence interval (97.5%)
Preconceptionally after surgery	0.125	0.329	0.781	–0.518
First trimester	1.074	0.343	1.743	0.404
Second trimester	0.919	0.343	1.591	0.246
Third trimester	0.424	0.437	1.281	–0.432
**Vitamin D**				
	Beta	Standard error	Lower confidence interval (2.5%)	Upper confidence interval (97.5%)
Preconceptionally after surgery	7.941	3.657	0.774	15.108
First trimester	6.519	3.501	–0.343	13.381
Second trimester	8.659	3.719	1.369	15.948
Third trimester	11.302	4.116	3.234	19.369
**Ferritin**				
	Beta	Standard error	Lower confidence interval (2.5%)	Upper confidence interval (97.5%)
Preconceptionally after surgery	15.952	11.640	–6.862	38.766
First trimester	29.430	12.726	4.488	54.373
Second trimester	23.572	12.776	–1.470	48.614
Third trimester	7.846	13.825	–19.250	34.943

* All micronutrient values represent absolute differences, whereas vitamin B12 values represent relative differences. For vitamin B12, a difference of 0.05 corresponds with a difference of 5%. Linear mixed models were used, adjusted for season, ferric carboxymaltose infusions, vitamin B12 injections, diabetes mellitus, ethnicity, smoking, educational level, age, time interval between blood draw and bariatric surgery, first vs. consecutive pregnancy, and BMI at blood draw.

## Data Availability

The data presented in this study are available on request from the corresponding author.

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
