# Peer review of "The Impact of Preconception Gastric Bypass Surgery on Maternal Micronutrient Status before and during Pregnancy: A Retrospective Cohort Study in the Netherlands between 2009 and 2019"

_nutrients, 2022, doi:10.3390/nu14040736_

Round 1
Reviewer 1 Report
Review of nutrients-1541267-v1
This is a potentially interesting paper about an interesting, if niche, subject.
I note that I am the statistical reviewer, with some experience in studies of pregnancy and nutrients, but all in low and middle income countries, so please forgive any misunderstandings I have of the situation discussed in this paper.
As a general principle, I believe that all papers should indicate something about where and when the research was conducted, both in the title and in the titles or footnotes of all tables and figures. Place and time (of the comparator papers) are also relevant when discussing the relationship of a paper to the existing literature (Introduction and Discussion sections). Many epidemiological studies are highly context dependent. In addition, all tables showing the results of analyses need some short indication (e.g., briefly in a footnote) describing the analytic method used.
I’m stepping out of line here, but it seems to me that in nutrition, and in medicine in general, there are “normal” ranges that are set by “what is,” and those that are set by “what is necessary/good for people?” An example of the former is the fact that the low end of normal hemoglobin levels is set differently for men and premenopausal women. An example of the latter (I think) is that criteria for polycythemia are set based on “what is bad for people.” Similarly, blood pressure and blood sugar level cutoffs (in general and during pregnancy) are based on their relationship to pathology. In addition, there are often rules of thumb for nutrient intake like “add 10% for pregnant women,” and I can imagine something similar for blood levels, though perhaps the intake increases are meant to maintain the blood levels. All this is a propos the great variation seen in the “normal” boundaries for hemoglobin, folate (where I suspect that the higher levels in the second and third trimesters reflect intake of prenatal vitamins), vitamin B12, vitamin D, and calcium. Now that I have looked at the graphs, I am not at all sure that I care whether someone’s vitamin B12 is categorized as high during pregnancy based on the criteria used. This all makes me think that I’d much rather see an analysis of the continuous values. Most of the values categorized as high are below the upper limit of the pre-pregnancy “normal,” though obviously a few seem excessive.
I wrote most of this review taking the authors’ categorizations as given (and I’m leaving it as originally written), but, as I said in the previous paragraph, I’m not sure I’m happy about that. As I said, this is not my “territory,” so I’m happy to bow to the editors’ judgment and to entertain reasonable arguments for these different sets of limits by time.
Please take special notice of my remarks below about analysis methods, as well as other analyses that could be done with these data. I suspect that the underlying data are likely to be high quality and not so easy to get, so full use should be made of them (though not necessarily all in one paper).
Specific remarks:
Partial paragraphs at the top of a page are counted. The SECOND paragraph on page 2 begins with “Bariatric.”
Page para- line comment
graph
2 2 5-6 Please indicate where and when these values come from, as in “in (country X) in (year Y), 73.7%.....” One could imagine that numbers elsewhere and in other times would be different. The most appropriate numbers would be from their clinic.
3 Did all the pregnancies result in live births?
The authors tell us that there are multiple types of BS, but then seem to treat all their patients/subjects as having had GB. Please indicate in the Methods section why this is so (e.g. special code for GB, or their center only does GB, or they excluded patients with other types of surgery, and if so, how many?). Is a figure appropriate/needed for this information?
4 1 Given the long period over which these data were collected, it seems likely that they were routinely collected as part of clinical practice. If this is so, please say so. If it is not, please explain. Were frozen blood specimens used? Were there really no missing data? If there were missing data, what did the authors do about them?
3 The authors do not describe any analysis taking repeated measures on patients (including before and during each pregnancy, and multiple pregnancies of individuals) into account. It might be useful to use generalized estimating equations (GEE). Treating the measurements as independent is wrong, as it artificially increases the number of degrees of freedom, narrows the confidence intervals, and lowers the p-values. Nor do they describe any multivariate techniques that might improve the validity and understanding of the different effects. They say they want to describe, but once they start doing statistical tests, they might as well get the most mileage (I’m American) from their data. Cross-sectional relationships between the outcomes are also of interest (e.g., hemoglobin, iron, vitamin B12 (and maybe others of which I’m unaware)).
It is also of interest to know what the “reliability” (i.e. intraclass correlation coefficient, ICC) of the micronutrient values is over the course of time in this study. Do values in an individual tend to stay in the “same” level, or do they jump around? Are the women who had high (or low) values at some time during the first pregnancy the same as those with these problems during the second pregnancy (if they had one)? This is an important question for clinical practice. If I were a patient and the current regimen left me with chronic deficiencies or excesses that could have adverse consequences, I would want my doctor to change the regimen to be more appropriate for me. But if each person’s values show no consistency, there is no way to adjust the multivitamin regimen.
If there was a prespecified analysis plan for this study, the results from analyzing different sets of variables, as I have suggested above, could be written up as exploratory, or shown without p-values.
If a proper analysis is beyond the capabilities of the current set of authors, please consult a statistician. I’m sure there are good ones in the Netherlands.
6 Is it possible that patient and environmental characteristics (e.g., darker skin in a northern latitude, season) affect vitamin D levels? Similar questions can probably be asked about other micronutrients.
7 Hemoglobin decreased while iron increased. So most anemia found is not iron deficiency anemia? Since these data are from a clinical context, was there any attempt to determine the causes of anemia in the presence of normal or higher iron? Please show all the p-values and note that analysis was by chi squared test in a footnote.
Table 1. Suggested title: Baseline….included post-gastric bypass……), Netherlands, 2009-2019
The data are based on medical records. How can they not know the ages of 9 people (at least to some useful level)?
“apnea syndrome” is the only item without an initial capital letter.
Other/mixed is a large category for the Netherlands, it seems to me.
Table 2. Why are the values in the line Hemoglobin/T1 bolded?
Table 3. They treat the numbers in this table as a set of independent comparisons of FFM vs. standard multivitamins within trimester and level. This is clearly wrong, as the sum of the proportions in the 3 levels is always 1. THE MINIMUM acceptable level of analysis would be a 3x2 table of level by type of multivitamin within trimester. And that ignores the likely clustering within women.
Whatever the analysis is done (short of GEE), it is necessary to correct for multiple testing, or, if the journal permits, to state that no adjustment was made for multiple testing.
The number of subjects in this table is 145. Are the other 52 (of 197) second pregnancies? “impure” multivitamin regimens (e.g., with injections)? This deserves at least a footnote here, as well as description in the Methods section.
Fig. 2 Why are some of the circles in the Hemoglobin graph, first trimester, black (symbolizing before surgery)?
Also, I find the horizontal axis confusing. It seems that the first section is in days after surgery (but that’s not so clear, and it’s labeled weeks, but 400 weeks is over 10 years), while the other three sections are shown as weeks of gestation.
For folate, there seem to be a lot of values along a line at the top. Is this the upper limit of the assay?
Whatever the authors do, they should label the graphs carefully and informatively, either on the graph and its axes or in the figure legend.
I think that at this stage it is relatively pointless for me to review the Discussion section.
Suggested model with continuous outcomes:
<micronutrient value> ~ <time since GB> <age at conception> <parity> <smoking> <comorbidity> <BMI at conception> <geographic origin> <season> <multivitamin regimen> <trimester (with a code for pre-pregnancy)> <pregnancy number since GB>
with patient as the grouping/cluster variable.
It is reasonable to test (in some relatively generous way) all the variables other than <multivitamin regimen> and trimester for inclusion in each model (i.e. different models for different outcomes). Variables can be used as linear continuous (though this can be tricky sometimes), or categorized, as appropriate (e.g., time since GB: under 18 months vs. at least 18 months; pregnancy number: 1 or greater than 1).
It may also be possible to make models of the same sort with categorical/multinomial outcomes, though their power will be much lower than that for continuous outcomes. Also, see above for my misgivings about the categories. I suppose that the predicted values from the continuous models could be categorized, though I’m not so sure how helpful this will be.
It is conceivable that there will be some significant interactions, though with this sample size, the power to detect them is low.
Reviewer 2 Report
1/ This is an interesting retrospective observational study in a cohort of 197 singleton pregnancies following gastric bypass surgery with the aim to check how the procedure affects hemoglobin, mineral and vitamin maternal status before and during pregnancy. Importance of such data and findings is obvious in the era of pandemic obesity in current reproductive population.
2/ It is not cleared why only one of three available surgical procedures was applied in all patients. What is the reason?
3/ Since the authors are Ob/Gyn we could expect more perinatal data concerning pregnancy complications at least mentioned in the manuscript like rate of miscarriage, preterm births, preeclampsia, fetal growth restriction/small for gestational age, gestational hypertension, GDM etc. as well as perinatal outcome like gestational age at delivery, mode of delivery e.g. in percentage, neonatal birth weight, neonatal gender, NICU admissions etc.
4/ It is not cleared what was the rate of fetal defects especially bound with vitamin status such as neural tube defects, heart defects, cleft lip and/or palate and other body central line defects and whether any correlation with deficits or excess was found. It would be valuable to check separately malformation rate in deficit subgroup and excess one.
5/ It is not cleared how many cases with overdosed levels of vit B12 or vit D3 were in toxic ranges and if there were any complications related to.
6/ Looking at graphs with lines of normal ranges we are surprised by vast differences between trimesters resulting in many cases being out of range in 1st trimester contrary to most being within ranges in 2nd or 3rd one. What could be the explanation for each of the studied factors?
7/ "Excess levels of vitamin B12 in 23.6-41.3% throughout all trimesters were noted, and vitamin excess levels were not related to vitamin B12 injections. 36.9% of the patients (n=69) were administered additional vitamin B12 injections during pregnancy." It is nor cleared what were the indications for extra vitB12 injections and how was it established that excess levels did not correlate with them
